# Development of a Rapid Detection Method for Ethylene Glycol and Glycolic Acid in Feline Samples: A Response to Increasing Antifreeze Poisoning Incidents in Korea

**DOI:** 10.3390/ijms252011030

**Published:** 2024-10-14

**Authors:** HyunYoung Chae, Jae Won Byun, Go-Eun Shin, Kyung Hyun Lee, Ah-Young Kim, Bok-Kyung Ku, Md Akil Hossain, Tae-Wan Kim, JeongWoo Kang

**Affiliations:** 1Animal Disease Diagnosis Division, Animal and Plant Quarantine Agency (APQA), Ministry of Agriculture, Food and Rural Affairs, 177, Hyeoksin 8-ro, Gimcheon-si 39660, Republic of Korea; iichy33ii@korea.kr (H.C.); jaewon8911@korea.kr (J.W.B.); tlsrhdms924@korea.kr (G.-E.S.); mylovehyun@korea.kr (K.H.L.); mochsha@korea.kr (A.-Y.K.); kubk@korea.kr (B.-K.K.); 2Laboratory of Veterinary Physiology, College of Veterinary Medicine, Kyungpook National University, Daegu 41566, Republic of Korea; twkim@knu.ac.kr; 3Institute for Tuberculosis Research, College of Pharmacy, University of Illinois Chicago, 833 S. Wood St. (MC964), Chicago, IL 60612, USA; mdakil@uic.edu

**Keywords:** antifreeze poisoning, ethylene glycol, glycolic acid, rapid diagnostic method

## Abstract

Recently, cases of antifreeze poisoning in companion animals, particularly cats, have surged in the Republic of Korea. Ethylene glycol (EG), the toxic primary component of antifreeze, is metabolized into glycolic acid (GA), leading to severe metabolic acidosis, acute kidney injury, and death. Traditional detection methods, although effective, are often time-consuming owing to complex sample preparation. This study involved a novel analytical method utilizing GC-MS for EG and LC-MS/MS for GA detection, which streamlined the detection process by eliminating the need for derivatization. The method was validated for accuracy and reliability, enabling the rapid and precise identification of EG and GA in biological samples. This study also included the successful application of this method in a case where initial exposure to antifreeze was not apparent, which highlighted the effectiveness of this method in diagnosing poisoning even in cases where clinical history is unclear. The development of this rapid diagnostic approach addresses the urgent need for the efficient detection of antifreeze poisoning, improving animal welfare and supporting forensic investigations.

## 1. Introduction

Recently, in the Republic of Korea, there has been a notable increase in cases of antifreeze poisoning among companion animals, particularly felines. Ethylene glycol (EG), a key component in antifreeze formulations, lowers the freezing points of automotive and industrial coolants [1,2]. Despite its utility and wide use, EG is colorless, odorless, and has a sweet taste that masks its highly toxic nature, leading to accidental or intentional ingestion and potentially fatal outcomes. In particular, the palatability of antifreeze poses a significant risk to animals [3]. However, cats may ingest antifreeze not because of its taste but due to their grooming habits, licking it off their fur if it adheres to them. The transformation of EG into glycolic acid (GA), a toxic metabolite, through metabolic processes contributes to various adverse effects, such as metabolic acidosis and acute kidney injury [4]. In our process, we have primarily relied on necropsy findings, including hematoxylin and eosin (H&E) staining of collected tissues and genetic testing, but there have been cases where the results were inconclusive, highlighting the need for a more precise diagnostic method. This diagnostic uncertainty poses immediate health implications, encompassing broader concerns regarding animal welfare and presenting substantial legal and ethical challenges. Therefore, there is an urgent need for reliable and rapid diagnostic methods capable of identifying EG and its metabolites in biological samples.

Traditional analytical methods, including gas chromatography–tandem mass spectrometry (GC-MS/MS) and liquid chromatography–tandem mass spectrometry (LC-MS/MS), have been instrumental in toxicological studies. However, these techniques often involve complex sample preparation and derivatization steps, such as the use of bis (trimethylsilyl)trifluoroacetamide (BSTFA) to enhance compound volatility and detection sensitivity [5]. Although effective, these processes are time-consuming and prone to non-specific reactions. Additionally, the amount of sample that can be obtained from companion animals is typically small, which presents a challenge to achieving accurate and reliable results. Moreover, the limited sample volume makes it difficult to test for other potential compounds that could be involved. Therefore, a streamlined analytical process that enables rapid and precise detection of EG and GA, while allowing assessment of other possible toxicants, is critically needed to facilitate prompt and appropriate medical intervention.

This study introduces a novel analytical methodology that obviates the need for derivatization, simplifies the detection process, and significantly reduces the likelihood of analytical errors [6]. This methodological innovation provides a dependable tool for rapid diagnosis of EG poisoning in companion animals, aiding in the timely administration of treatment and supporting legal action in cases of suspected animal abuse [7,8]. Broadly, our study addresses a pressing public health concern with wide implications for animal welfare, forensic science, and toxicology.

## 2. Results

### 2.1. Analysis of Ethylene Glycol Using GC-MS

A new method using GC-MS was developed for the accurate quantification of EG in the stomach tissues of felines. This approach involved the measurement of EG, with intense mass spectral peaks at m/z 62, 33, and 31. Based on these observations, the quantification ion was designated as 31 m/z, whereas the qualitative ions were set as m/z 33 and 62. The ion ratio (qualitative/quantitative) was 0.33. According to the standards, the range of quantifier/qualifier ratios allowed is usually less than ±30%. Before sample analyses, the procedure was validated using stomach tissue samples spiked with EG. The chromatograms produced (Figure 1) under the specific chromatographic conditions used in this study demonstrated no significant interference from the tissue matrix, thereby ensuring reliability of the results. EG had a retention time of approximately 9.15 min. The method exhibited a linear response over the concentration range required for the objectives of the research, with a correlation coefficient (*R*^2^) of 0.99 at concentrations ranging from 5 to 200 μg/mL in the tissue. The limits of detection (LOD) and quantification (LOQ) were 25 and 100 ng/mL, respectively. Recovery and precision of EG using this method were in the range of 83.05–91.73% (Table 1). These results validated the repeatability and accuracy of the method for EG analysis.

### 2.2. Analysis of Glycolic Acid Using LC-MS/MS

An analytical approach utilizing LC-MS/MS was used to quantify GA, a key metabolite of EG, in feline gastric tissues [9]. Mass spectral peaks were identified at 74.9 m/z as the precursor ion, accompanied by product ions at 45.1 and 35.0 m/z. Based on these observations, the quantification ion was designated as 45.1 m/z, whereas the ion chosen for qualitative analysis was set as 35.0 m/z. The ratio of qualitative to quantitative ions was set to 0.5. To ensure the validity of the method, stomach tissue samples spiked with GA were analyzed before the technique was used for actual sample evaluation. The resulting chromatograms (Figure 2) from this methodology highlight the absence of significant matrix-related interference, confirming the capacity to yield dependable outcomes for the measurement of biological GA levels. Under the chromatographic conditions specified above, GA exhibited a retention time of 1.10 min. This approach demonstrated a linear relationship across the range of concentrations deemed necessary to achieve the goals of this study. The correlation coefficient (*R*^2^), LOD, and LOQ were 0.99, 5, and 20 ng/mL, respectively. The recovery rate using this method fell within 80.92–95.30%, and the precision estimates expressed as relative standard deviations (RSDs) for GA concentrations of 1, 2, 5, 25, and 50 μg/mL in the tissues were consistently 0.81–1.34% and 0.41–0.96% for GA and EG, respectively (Table 1). These findings confirmed the reliability and accuracy of the method within a single analytical session.

### 2.3. Analysis of Samples Suspected to Be Exposed to Antifreeze

A stray cat (case 1) was admitted to a veterinary hospital exhibiting symptoms of vomiting, respiratory distress, and decreased consciousness. Despite emergency interventions, the cat did not survive. A toxicology analysis was requested from the Animal and Plant Quarantine Agency to determine the cause of death. Upon necropsy, kidney lesions were observed, and histopathological examination using H&E staining confirmed the presence of crystalline deposits of translucent material in the renal tissue (Figure 3). However, the specific nature of the crystals could not be identified. Biomarker testing using reverse transcription–quantitative polymerase chain reaction (RT-qPCR) for markers typically associated with EG exposure was conducted, but no definitive results were obtained.

The presence of EG and GA in the stomach sample was confirmed using the mass spectrometry method developed in this study; 82 µg/mL EG and 27 µg/mL GA were detected in case 1. Subsequently, four additional cases (case 2–5) of suspected EG exposure were identified, and calcium oxalate crystals were observed in the kidney tissue upon necropsy. In total, five cases of EG poisoning were documented, with significant amounts of EG and GA detected. The concentrations of EG and GA in the samples exceeded the measurement capabilities of GC-MS/MS and LC-MS/MS. To accurately quantify these high concentrations and establish the ratios of the detected EG and GA ions, the samples were diluted 1000-fold for re-analysis (Figure 4). The detected concentrations were 82–590 μg/mL for EG and 27–98 μg/mL for GA in the stomach tissues.

As EG is metabolized in the liver, and the primary lesions appeared in the kidneys, additional liver and kidney samples were collected from the cases for further analysis. The results are presented in Table 2. In the liver, EG and GA concentrations were 69–473 μg/mL and 29–62 μg/mL, respectively, whereas in the kidneys, EG and GA concentrations were 67–410 μg/mL and 28–55 μg/mL, respectively.

## 3. Discussion

EG and its toxic metabolite GA are of significant concern in veterinary forensics owing to their high toxicity and the frequent occurrence of antifreeze poisoning in pets [10]. The need for precise diagnostic methods is underscored by the palatable taste of ethylene glycol-containing antifreeze, which often leads to accidental ingestion by companion animals and poses severe toxicological risks, resulting in potentially fatal outcomes [11]. Upon ingestion, EG is metabolized to glycolaldehyde by alcohol dehydrogenase, which is subsequently oxidized to GA. This metabolite is further converted to oxalic acid, which can form calcium oxalate crystals in the kidneys and cause renal damage [12,13]. Recent incidents in Korea have highlighted the fatal consequences of antifreeze ingestion, with necropsies revealing gastric contents indicative of acute poisoning. Moreover, visual observations of renal tubular epithelial necrosis accompanied by calcium oxalate crystals underscore the severe and rapid progression of toxicity associated with EG exposure [14].

In one notable case (case 1), the initial necropsy revealed no evidence of antifreeze exposure, and the presence of bleeding led to the suspicion of rodenticide poisoning. However, subsequent tests did not support this hypothesis. Further investigation of kidney tissues using H&E staining identified mild crystalline deposits, raising the possibility of antifreeze poisoning. To confirm this, molecular testing was performed, since microRNA expressed in renal tissue is typically responsive to EG exposure. However, the biomarker tests showed no significant changes, likely owing to the low sensitivity of the assay. To accurately determine the cause of death, a mass spectrometry method was developed. This novel approach enabled the precise detection and quantification of EG and its metabolites for veterinary forensic applications, particularly concerning companion animals like cats, ultimately confirming antifreeze poisoning as the cause.

Recently, a method utilizing GC-MS/MS with BSTFA derivatization has been reported for the precise measurement of EG and its metabolites in human samples [15]. However, this method is insufficient for veterinary forensic applications in cats for several reasons. Firstly, the amount of tissue available from feline necropsy is very limited—typically only 3–4 g—and the analytical procedures can consume about 1–2 g of sample, representing a substantial fraction of the available tissue. The derivatization process further consumes significant portions of these small samples, hindering the simultaneous detection of other potential toxicants. In cases of sudden animal death, minimal prior information is often available, making it necessary to perform comprehensive screening for a wide range of toxic substances, including pesticides and drugs, unless EG poisoning is certain. Secondly, the extent to which the recently reported method is affected by the matrix effects specific to cat tissues has not been confirmed. To overcome these challenges, we developed a novel method specifically tailored for veterinary forensic applications, using GC-MS for EG and LC-MS/MS for GA without the need for BSTFA derivatization. This method was designed to rapidly detect and accurately quantify EG and GA in the same sample, even when other toxic substances, such as rodenticides or pesticides, were present. The method met all validation parameters, with proven reproducibility and stability [16,17].

Our analysis of EG and GA concentrations in five feline cases of antifreeze poisoning revealed a correlation between the presence of these compounds in various tissues and pathological outcomes. For example, in case 1, EG was found in gastric tissue at a concentration of 82 μg/mL and in renal tissue at 67 μg/mL, with corresponding GA concentrations of 27 μg/mL in gastric and 28 μg/mL in renal tissues. These concentrations suggest that exposure at this level leads to organ bleeding and formation of mild calcium oxalate crystals in the kidneys, indicating renal damage. Similarly, in case 2, higher levels of EG and GA were detected, with EG at 572 μg/mL in gastric tissue and 367 μg/mL in renal tissue, with GA at 59 μg/mL in gastric and 46 μg/mL in renal tissues. The presence of calcium oxalate crystals in the kidneys supported the diagnosis of antifreeze poisoning. This pattern of EG metabolism to GA and its subsequent conversion to oxalic acid, which forms calcium oxalate crystals, was observed in all cases. For instance, in case 3, the levels of EG were 590 μg/mL in gastric tissue and 359 μg/mL in renal tissue, whereas GA levels reached 98 μg/mL in gastric and 53 μg/mL in renal tissues, again correlating with the presence of calcium oxalate crystals in the kidneys. In cases 4 and 5, similar patterns were observed, as seen in the previous cases.

This variability in the metabolic response, combined with the consistent presence of calcium oxalate crystals across cases, underscores the deadly nature of EG poisoning and critical need for rapid diagnosis and treatment. These findings support forensic analyses of metabolites in cases of suspected antifreeze ingestion and demonstrate that even low concentrations of EG and GA can lead to fatal outcomes. The potential lethality of EG and its metabolites highlights the urgent need for rapid diagnostic methods and effective treatment strategies to mitigate the risks associated with antifreeze poisoning in pets. The presence of calcium oxalate crystals serves as a definitive indicator of toxicity progression, aiding in the forensic reconstruction of events leading to poisoning, which is vital for both therapeutic and legal outcomes.

## 4. Materials and Methods

### 4.1. Reagents and Chemicals

High-purity EG and GA were purchased from Sigma-Aldrich (St. Louis, MO, USA) and used as standards. LC-MS-grade acetonitrile and methanol were purchased from J.T. Baker (Phillipsburg, NJ, USA), and formic acid solution was obtained from Merck (Darmstadt, Germany). Distilled water (EMD Millipore, Billerica, MA, USA) was used with a Milli-Q Integral Milli-Q membrane point-of-use cartridge (0.22 μm). All other reagents and solvents were of analytical grade.

### 4.2. Sample Preparation for Mass Spectrometry Analysis

Upon receiving a report of a deceased feline, a necropsy was conducted to remove the maximum amount of stomach content or tissue. The samples were placed in a container designed for freezing for preservation. For the analysis, 2 g of the sample and 4 mL of cold acetonitrile with 0.1% formic acid were added to a conical tube. The supernatant obtained by centrifugation for 10 min at 3000× *g* and 4 °C was put into an ultracentrifuge filter (cutoff membrane at 10 kDa, 4 mL) and purified by centrifugal separation (Beckman Coulter Avanti J-E; 3000× *g*, 4 °C, 2 h). The purified solution was placed in a nitrogen evaporator, concentrated to 1 mL, and filtered using a 0.2 μm PVDF syringeless filter prior to analyses. The samples were divided equally into two groups. One half (0.5 mL) was used to measure EG by GC-MS. The remaining 0.5 mL was utilized for the quantification of GA using high-performance LC-MS/MS.

### 4.3. GC-MS Method

EG was analyzed by GC-MS (GCMS-TQ8050 NX, Shimadzu, Kyoto, Japan). The instrument was equipped with an SH-PolarWax column (0.32 mm ID, 1 µm, 30 m) to achieve high-resolution separation. The analytical conditions were optimized as follows: The injection volume was set at 1 μL and injection temperature was maintained at 250 °C. The column flow was regulated at 2.0 mL/min in column flow mode, with a split injection mode of 5:1 to ensure optimal sample introduction. The column oven temperature program started at 60 °C, was held for 2 min, and was then ramped up at a rate of 15 °C/min to 240 °C, where it was held for an additional 3 min. The ion source and interface temperatures were set at 200 °C and 230 °C, respectively. EG was quantified in selected ion monitoring (SIM) mode, targeting ions with m/z ratios of 31, 33, and 62 for high specificity and sensitivity.

### 4.4. LC-MS/MS Method

GA in the solution was quantified using a SCIEX Triple Quad 5500+ system (AB Sciex, Framingham, MA, USA) equipped with an electrospray ionization (ESI) interface for ion generation. Chromatographic separation was achieved on a Phenomenex LC column (2 × 50 mm, 4 μm particle size maintained at a temperature of 40 °C). The flow rate was set to 0.2 mL/min and the injection volume was 2 μL. The mobile phase consisted of two components: 5 mM ammonium acetate in distilled water (mobile phase A) and 0.1% formic acid in methanol (mobile phase B). A gradient program was applied to optimize the separation with a total run time of 8 min. The program was initiated with 50% B for 2.0 min, increased to 95% B from 2.5 to 5.0 min, and then returned to 50% B from 5.1 to 8.0 min. The analytes were ionized in negative mode using an ESI source. For the quantification of GA, multiple reaction monitoring (MRM) was employed, focusing on the transitions 74.9, 47.0, 45.1, and 35.0 to ensure precise and accurate measurements.

### 4.5. Validation of GC-MS and LC-MS/MS Methods

To assess the specificity of the quantification methods for detecting EG and GA in tissue samples, standard solutions of these compounds were prepared and used to spike untreated tissue samples with concentrations of 10, 50, 100, 500, and 1000 ng/mL. This procedure ensured that matrix effects did not interfere with detection. The standard solutions were prepared in acetonitrile with 0.1% formic acid, and the pH was adjusted to maintain stability and ensure a linear response. Dilutions were performed to create calibration standards. The samples were analyzed using GC-MS and LC-MS/MS to determine the concentrations of EG and GA. The collected data were used to evaluate accuracy, linearity, recovery percentage, calibration curves, and regression coefficients. The consistency and reliability of the method were verified by injecting samples at three different concentrations, six times each. The limits of detection (LOD) and quantification (LOQ) were calculated from the calibration curve using the equations LOD = (3.3 × SD)/slope and LOQ = (10 × SD)/slope, where SD denotes standard deviation of signal responses [18].

### 4.6. Forensic Examination [Necropsy and H&E Staining]

Upon receiving samples from deceased animals suspected of EG exposure, a detailed necropsy was performed. Sample selection was based on the observation of specific pathological signs indicative of EG toxicity, such as renal tubular epithelial necrosis accompanied by calcium oxalate crystals, pulmonary edema, and hemorrhagic gastroenteritis [19,20]. H&E staining of kidney tissues was performed after fixation, processing, embedding, and sectioning using a Ventana HE 600 (Roche Diagnostics, Tucson, AZ, USA), following an in-house method. Five cases exhibiting these characteristics were identified. Stomach tissue samples were collected to test for EG and GA.

### 4.7. Biomarker Test Using PCR

The expression of differentially expressed microRNAs (miRNAs) after antifreeze poisoning was verified using RT-qPCR. Nucleic acid was extracted from the kidney tissue using the miRNeasy Mini kit (Qiagen, Hilden, Germany), according to the manufacturer’s instructions. RT-qPCR was conducted using specific primer sets based on a previous study [21].

## Figures and Tables

**Figure 1 ijms-25-11030-f001:**
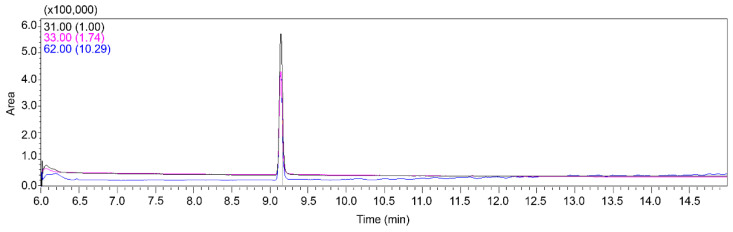
Mass chromatogram of ethylene glycol in feline gastric tissue obtained by GC-MS.

**Figure 2 ijms-25-11030-f002:**
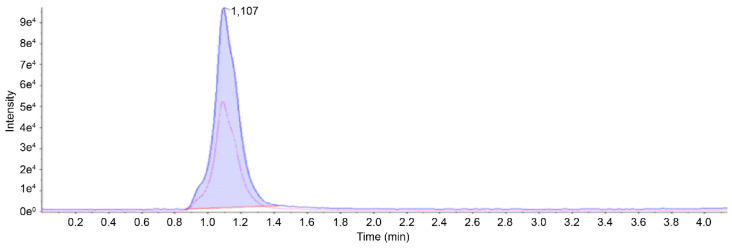
Mass chromatogram of glycolic acid (precursor and daughter ion) in feline gastric tissue obtained by LC-MS/MS.

**Figure 3 ijms-25-11030-f003:**
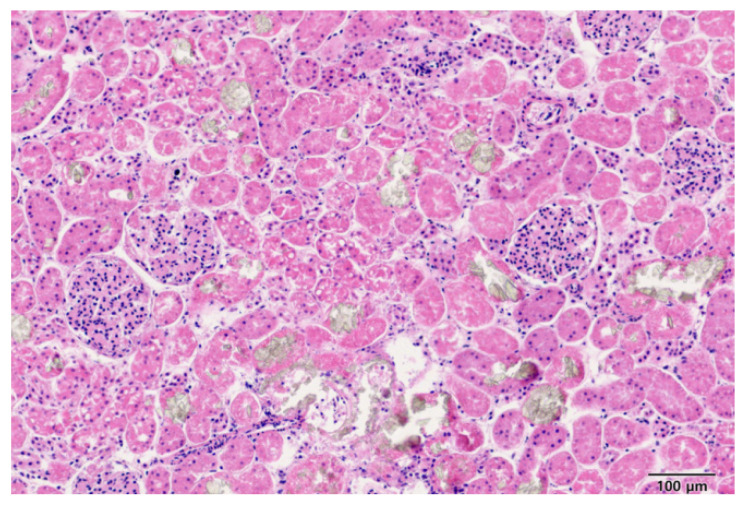
Deposition of salt components, visible as translucent crystals, in the renal tubular lumen, and resulting tubular epithelial damage in tissue sample from case 1.

**Figure 4 ijms-25-11030-f004:**
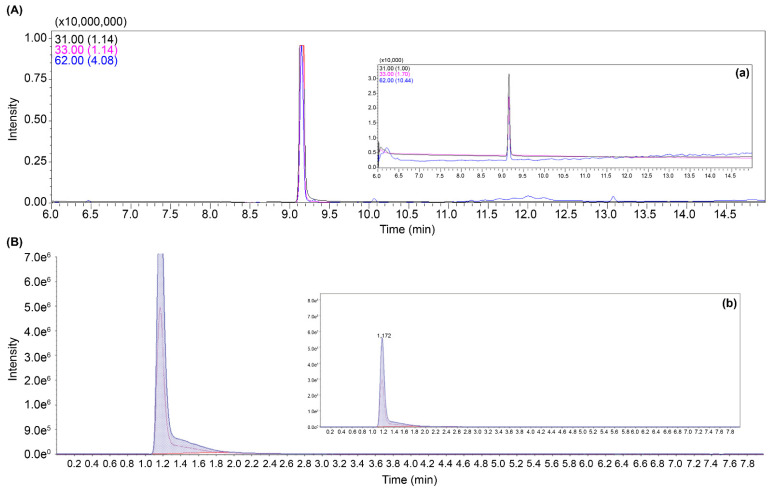
Mass chromatogram of EG (**A**), 1000-fold diluted EG (**a**), GA (**B**), and 1000-fold diluted GA (**b**) in feline gastric tissue sample from case 3.

**Table 1 ijms-25-11030-t001:** Validation parameters for ethylene glycol and glycolic acid detection using GC-MS and LC-MS/MS methods.

Compound Name	Calibration Curve (ng/mL)	r^2^	LOD (ng/mL)	LOQ (ng/mL)	Recovery (%)	RSD (%)
Ethylene glycol	50–1000	0.99	25	100	83.05–91.73	0.41–0.96
Glycolic acid	10–200	0.99	5	20	80.92–95.30	0.81–1.34

**Table 2 ijms-25-11030-t002:** Concentrations of ethylene glycol and glycolic acid in samples from fatal feline cases measured using the LC-MS/MS and GC-MS methods.

Case	Sample Characteristics	Analytes	Concentration (µg/mL)			Other Relevant Findings
			Gastric Tissue	Liver Tissue	RENAL TISSUE	
**1**	Age: - Sex: Male Weight: 3.23 kg Location: Suwon	EG	82	69	67	Upper GI tract bleeding, mild calcium oxalate crystals within the kidneys
		GA	27	29	28	
**2**	Age: - Sex: Female Weight: 4.8 kg Location: Seoul	EG	572	395	367	Calcium oxalate crystals within the kidneys
		GA	59	41	46	
**3**	Age: - Sex: Female, Weight: 4.6 kg Location: Seoul	EG	590	473	359	Calcium oxalate crystals within the kidneys
		GA	98	62	53	
**4**	Age: - Sex: Male Weight: 4.2 kg Location: Seoul	EG	541	313	410	Calcium oxalate crystals within the kidneys
		GA	43	50	55	
**5**	Age: - Sex: Female Weight: 3.1 kg Location: Gimcheon	EG	497	387	340	Calcium oxalate crystals within the kidneys
		GA	40	51	42	

## Data Availability

The data presented in this study are available in the article.

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
