# Peer review of "Development of a Rapid Detection Method for Ethylene Glycol and Glycolic Acid in Feline Samples: A Response to Increasing Antifreeze Poisoning Incidents in Korea"

_ijms, 2024, doi:10.3390/ijms252011030_

Round 1
Reviewer 1 Report
Comments and Suggestions for Authors
This is interesting and although I cannot comment on the details of the method I can comment on the clinical aspects of EG poisoning in pets.
Lines 18-19
…leading to severe conditions such as metabolic acidosis and acute renal failure.
This would be better as …leading to severe metabolic acidosis, acute kidney injury and death.
Line 32
Why has there been an increase in EG poisonings in Korea? Is it a real increase or has diagnosis or laboratory confirmation increased? Are there any figures for cases per year or cases confirmed by the APQA per year?
Line 38
Reference 3 does not mention palatability but says cats and dogs drink EG because of its sweet taste. This is not true as cats cannot taste sweetness. Perhaps ‘apparent palatability’ is better. IT may also be a question of volume. Cats care particularly susceptible to EG, although it not clear why.
Line 40
Replace acute renal failure with acute kidney injury.
Line 40
Is it worth mentioning that cats are at particular risk of EG toxicosis? They have a small body size compared to many dogs, and although they are more discriminating in their eating habits only a few millilitres can be fatal.
The lethal dose of ethylene glycol in cats is commonly reported as 1.5 ml/kg (Milles, 1946). In another study 1 g/kg was fatal to cats within 48 hours (Gessner et al., 1961).
Milles G. Ethylene glycol poisoning; with suggestions for its treatment as oxalate poisoning. Arch Pathol (Chic). 1946 Jun;41:631-8.
GESSNER PK, PARKE DV, WILLIAMS RT. Studies in detoxication. 86. The metabolism of 14C-labelled ethylene glycol. Biochem J. 1961 Jun;79(3):482-9. https://pubmed.ncbi.nlm.nih.gov/13704828/
Line 42
I would add a few words explaining what the ‘genetic’ testing is here. Are they better referred to as biomarkers?
..and biomarker testing (microRNA expressed in renal tissue)…
Line 55
‘typically small’ sample size.
Another factor is that cats often present late with EG toxicosis and the EG (which is not in itself toxic) has already been metabolised to toxic metabolites.
Line 140
Title of Table 2. Would be useful to add the sample are from fatal feline cases.
Concentrations of ethylene glycol and glycolic acid in samples from fatal feline cases measured using the LC- 140 MS/MS and GC-MS methods.
Line 150
Reference 11 – is it the correct reference? It is a human medicine reference.
Line 153
Replace acute renal failure with acute kidney injury.
Line 162
‘targeting genes’ Is this the correct terminology? Isn’t the testing for biomarkers (miRNA in this case) of EG toxicosis?
Table 1
What is RSD here? Relative standard deviation?
Figure 3.
Is this sample from Case 1?
Figure 4.
Are these results from a particular case?
Reviewer 2 Report
Comments and Suggestions for Authors
In this study, the authors tried to develop a novel analytical methodology that obviates the need for derivatization, simplifies the detection process, and significantly reduces the likelihood of analytical errors for ethylene glycol (EG) and glycolic acid (GA) in feline samples. The authors concluded that this variability in the metabolic response, combined with the consistent presence of calcium oxalate crystals across cases, underscores the deadly nature of EG poisoning and critical need for rapid diagnosis and treatment.
Comments:
The reviewer has some concerns as follows:
1. In this study, the authors showed a novel analytical method utilizing GC-MS for EG and LC-MS/MS for GA detection, which streamlined the detection process by eliminating the need for derivatization. However, their findings of the novel analytical method utilizing GC-MS are, in fact, similar to those revealed in a recent study. Tusiewicz et al. (2024) have revealed a novel technique for simultaneous ethylene glycol and its metabolites determination in human whole blood and urine samples using GC-QqQ-MS/MS. (J Xenobiot. 2024 Aug 27;14(3):1143-1164.). Therefore, the novelty for the analytical methods revealed in this study is really not high.
2. Can the deposition of salt components, visible as brown-colored material, also exist in the liver and gastric tissues excepting kidney? The histological analysis for the gastric, liver, and renal tissues needs to be shown.
3. The detailed information for feline samples can be described, such as age, sex, body weight, and others. Moreover, what is the source of these feline samples?
4. In Table 2, why samples from case 1 have the lowest levels of EG and GA? Their concentrations are very different from other cases. It can be explained and discussed.
5. The cited references can be updated with the recent studies.
6. Overall, the novelty and data presentation of this study are the major concerns, causing that the presented results cannot support the conclusions.
Round 2
Reviewer 2 Report
Comments and Suggestions for Authors
This revised manuscript has a great improvement. However, the novelty for this study needs to be further explained and discussed. The authors are recommended to discuss the issue of reviewer's previous comment #1 in the Discussion section.
Author Response
Comments to the Author: This revised manuscript has a great improvement. However, the novelty for this study needs to be further explained and discussed. The authors are recommended to discuss the issue of reviewer's previous comment #1 in the Discussion section.
Response: We sincerely thank you for taking the time to provide detailed and constructive feedback on our manuscript. Your suggestions have been invaluable in refining and enhancing the quality of our research. Accordingly, we have carefully considered your comments and incorporated additional explanations of the study's novelty, as well as addressed Comment #1 in the Discussion section. To this end, we have revised the Discussion to more clearly explain our novel approach and the unique solutions it offers in the forensic analysis of small animals. Furthermore, based on your feedback, we have emphasized the strengths of our analytical method, which is specifically designed for companion animals, particularly cats, thereby improving the manuscript's clarity and logical flow. We sincerely hope that these revisions meet your expectations. We are grateful for the opportunity to improve our research through your valuable insights, and we will continue to strive to further enhance the quality of our work. Thank you once again for your review, and we look forward to your positive response.
